# Semi-Supervised Learning for Forest Fire Segmentation Using UAV Imagery

**Junling Wang [1], Xijian Fan [1,\*], Xubing Yang [1], Tardi Tjahjadi [2] and Yupeng Wang [1]**

1   College of Information Science and Technology, Nanjing Forestry University, Nanjing 210037, China
2   School of Engineering, University of Warwick, Coventry CV4 7AL, UK
\*   Correspondence: xijian.fan@njfu.edu.cn

**Abstract:** Unmanned aerial vehicles (UAVs) are an efficient tool for monitoring forest fire due to its advantages, e.g., cost-saving, lightweight, flexible, etc. Semantic segmentation can provide a model aircraft to rapidly and accurately determine the location of a forest fire. However, training a semantic segmentation model requires a large number of labeled images, which is labor-intensive and time-consuming to generate. To address the lack of labeled images, we propose, in this paper, a semi-supervised learning-based segmentation network, SemiFSNet. By taking into account the unique characteristics of UAV-acquired imagery of forest fire, the proposed method first uses occlusion-aware data augmentation for labeled data to increase the robustness of the trained model. In SemiFSNet, a dynamic encoder network replaces the ordinary convolution with dynamic convolution, thus enabling the learned feature to better represent the fire feature with varying size and shape. To mitigate the impact of complex scene background, we also propose a feature refinement module by integrating an attention mechanism to highlight the salient feature information, thus improving the performance of the segmentation network. Additionally, consistency regularization is introduced to exploit the rich information that unlabeled data contain, thus aiding the semi-supervised learning. To validate the effectiveness of the proposed method, extensive experiments were conducted on the Flame dataset and Corsican dataset. The experimental results show that the proposed model outperforms state-of-the-art methods and is competitive to its fully supervised learning counterpart.

**Keywords:** forest fire monitoring; semi-supervised learning; semantic segmentation; convolution neural network

## 1. Introduction

Forest fire causes devastating disasters that are paroxysmal and uncontrollable. It not only reduces the forest stock, but also has a severe impact on forest growth. In addition, it is challenging to restore forests after a fire to their original state where the vegetation structure, species diversity and the ecosystem require a long time to recover [1]. Thus, it is of vital importance to monitor and detect forest fire accurately and rapidly [2].

There are currently four approaches to forest fire monitoring [3]: (1) ground patrols; (2) lookout station spotting; (3) satellite monitoring; and (4) unmanned aerial vehicle (UAV) patrols. Ground patrols are usually undertaken by forest rangers on foot or motorbike to check and monitor the implementation of rules and regulations for the prevention of forest fire, and to put out fire promptly when detected. This approach requires a large number of specialist staff and yet it is difficult to prevent forest fire. Lookout station spotting is usually located on the summit of a mountain or other high vantage points to spot a fire and determine its location. However, vegetation with varying size often obscures the spotting view. Satellite-based monitoring provides an efficient and large-scale wildfire assessment due to its wide range, time frequency and accuracy. However, this approach suffers from aerosol and cloud occlusions, which renders forest fire hardly detectable. Compared to satellite monitoring, UAV monitoring is undertaken at low-to-medium altitudes, which

effectively avoids the cloud interference [4]. It also has some advantages, e.g., cost-saving, lightweight, flexible and easily manipulated. Therefore, UAV has widely been used to monitor forest fire in recent years [5].

To address the challenges (e.g., varying scene illumination and complex back-ground) in monitoring forest fire using UAV imagery, a few studies involving machine learning have been undertaken. Ko et al. introduced a support vector machine (SVM)-based fire detection method, which removes non-fire regions based on brightness maps and designed a two-class SVM classifier with a radial basis function kernel [6]. Hossain et al. attempted to solve the problem of fire point monitoring via UAV images by using a single artificial neural network (ANN) [7]. Pérez-Rodríguez et al. used ANN-based classifiers to assess the feasibility of using multispectral images acquired on drones to estimate the severity of vegetation and soil after a forest fire [8]. However, traditional machine learning methods heavily rely on the effectiveness of feature extraction and the classifier, which lack the sufficient robustness and adaptability when applied to different environments.

In recent years, convolution neural network (CNN)-based methods have been introduced for monitoring forest fire due to its superior performance on computer vision tasks [9]. Most related studies leveraged the semantic segmentation methods to accurately identify the fire spots as well as its boundary [10,11] with promising results. However, all of these methods use fully supervised learning semantic segmentation and train the segmentation model with a large amount of pixel-wise annotated data. This approach is labor intensive and time-consuming. In fact, it is not trivial to obtain the pixel-wise annotation data for training CNN-based methods, especially in UAV images of forest fire. This is because forest fire tends to be of irregular shape and varying size, rendering it difficult to identify. The complex scene background may additionally confuse the forest fire recognition system, resulting in inaccurate annotation of the forest fire regions. Furthermore, manual annotation suffers from degradation due to artificial noise, which leads to the introduction of more useless information in the training model and thus decreases the performance of the model. Semi-supervised learning [12–14] is regarded as one of the promising techniques to deal with the above-mentioned problems, where a small amount of labeled data with a large amount of unlabeled data are mixed for model training, thus significantly reducing the manual annotations while increasing the performance of the model.

In this paper, we propose to segment forest fire in UAV-acquired images by using semi-supervised learning, where only a few pixel-wise labeled images are required for training the model. Our application of semi-supervised learning to forest fire segmentation provides a promising direction for the development of forest fire monitoring techniques. In addition, we take into account the three challenges exclusively present in fire segmentation in UAV imagery where the target fire could: (1) be partly occluded by vegetation; (2) have varying size and shape as well as boundary that is not easy to identify; (3) be distracted by complex scene background (e.g., soil, trees, snow, etc.). To this end, we focus on addressing the challenges by proposing a semi-supervised learning-based semantic segmentation network, SemiFSNet, for remote sensing forest fire monitoring. The major contributions of this paper are summarized as follows:

(1) A semi-supervised segmentation network specifically for forest fire segmentation in UAV optical imagery.

(2) A data augmentation strategy of random dropping a grid of image pixels, effectively alleviating the impact of occlusions caused by vegetation.

(3) A dynamic convolution module to encode multi-scale features in forest fire images extracted by fusing convolution kernels of different sizes, thus increasing the segmentation accuracy.

(4) A feature refinement module that cascades an attention module to encoded feature, thus highlighting the significant information of the target forest fire. This module further enhances the feature aggregation via two different types of attentions, thus enabling the model to focus more on forest fire while mitigating the influence of complex scene background.

(5) Extensive experiments are conducted on two publicly available datasets with different environments, demonstrating the promising performance of the model and the effectiveness of each contributing module of the proposed network.

The remainder of this paper is structured as follows. Section 2 reviews the related works and presents the proposed method. The experiment results are provided in Section 3. The discussion is given in Section 4 and the conclusions are drawn in Section 5.

## 2. Materials and Methods

### 2.1. Related Works

#### 2.1.1. Semantic Segmentation

Semantic segmentation aims to label each pixel of an image with a corresponding classification. Due to its superb performance, CNN-based image segmentation [15–18] has become the mainstream method. Long et al. [17] proposed full convolutional networks (FCN) by up-sampling (de-convolving) the output activation maps and fusing the output with the output of shallower layers. Badrinarayanan et al. [15] proposed an encoder-decoder segmentation network, SegNet, where the decoder is used to map the low-resolution feature representations to full-resolution feature maps for pixel-wise classification. As another encoder-decoder network, UNet was proposed by Ronneberger et al. [18], which involves a contracting path to extract context information and a symmetric expanding path to allow for precise localization. Chen et al. [16] proposed a segmentation architecture, DeepLabv3, which combines dilated convolutions with feature pyramid pooling. Chen2 et al. [19] extended DeepLabv3 by adding a simple yet effective refinement module in the decoder, namely, DeepLabv3+, which increases its capability in identifying object boundaries. In this paper, we use DeepLabv3+ as our basic encoder-decoder framework due to its two competitive advantages: (1) enabling to depict the multi-scale feature, which is widely existing in UAV-acquired forest fire images; and (2) significantly reducing the computational complexity, which is suitable for real applications.

#### 2.1.2. Semi-Supervised Semantic Segmentation

Traditional semantic segmentation methods have yielded promising performances, where the data used for their training are all pixel-wise annotated (referred to as full supervised learning). The acquisition of a large amount of annotated data tends to be time-consuming and labor-intensive, and the outcome of the training heavily relies on the quality of the labels used. To deal with the above-mentioned problems, semi-supervised learning has been proposed as an effective solution [20], where only a few labeled data are needed. Alternatively, many more unlabeled data are fully exploited to optimize the learning process as a complement, where unlabeled data are much easier to obtain when compared with the labeled ones [21]. The key problems in semi-supervised semantic segmentation are: (1) how to use the few labeled images; and (2) how to make full use of the many unlabeled images. Hong et al. [22] proposed to decouple classification from segmentation and used two separate networks for each task. The label of an image is predicted using the classification network, and each predicted label is binarily segmented using the segmentation network. The decoupled structure allows one to learn, classify and segment separately using the training data with image-level and pixel-wise class labels. Olsson et al. [23] proposed ClassMix data augmentation by blending unlabeled images, where the network predictions corresponding to target boundaries are leveraged to generate labeled bounding boxes. ClassMix data augmentation enriches the dataset by making full use of the unlabeled data. Adversarial learning-based semi-supervised segmentation is proposed in [24], where a full convolutional discriminator is used to distinguish the predicted probability maps from the ground truth distribution. Mondal et al. [25] proposed another adversarial learning-based method, where cycle consistency is used to learn bidirectional mapping between unpaired images generated by CycleGAN and segmentation masks. Yang et al. [26] proposed a self-training semi-supervised semantic segmentation by integrating data augmentation on unlabeled images. Lai et al. [27] proposed to retain the consistency related to context

information between features of the selected identity and accomplish consistency using directional contrastive loss, which achieves state-of-the-art performance. Hu et al. [28] proposed a new image synthesis and semi-supervised learning pipeline to train a site-specific weed detection model, enabling color matching between training and testing images, color enhancement during training, and iterative semi-supervised learning to greatly improve the performance of the model. Ke et al. [29] proposed guided collaborative training (GCT) for pixel-wise semi-supervised learning to learn additional information from the pseudo-segmentation generated by the model. However, they do not utilize generative models of the image itself, which limits the performance for the tasks requiring highly accurate pseudo-labels. French et al. [30] proposed that the network uses CutMix for semi-supervised learning and segmentation to achieve superior results on the PASCAL VOC 2012, CITYSCAPES, and ISIC 2017 datasets. However, for the aerial forest fire image, when the proportion of fire points in the whole image is small, CutMix will randomly erase a part of the pixel information on the image, which causes the reduction in the proportion of pixels containing information on the training image, thus resulting in the underfitting of the model. Although the semi-supervised segmentation has made substantive progress in recent years, only a few studies have investigated its application to remote sensing.

### 2.1.3. Semi-Supervised Semantic Segmentation for RS

There are a few studies exploring the application of semi-supervised learning to remote sensing. Yan et al. designed a semi-supervised method, which uses the generative adversarial network (GAN) [31] to enrich the training samples. With regard to semi-supervised semantic segmentation, Song and Yang [32] proposed the use of transfer learning and clustering for scene classification. Wang et al. [33] proposed the combination of consistency regularization and pseudo label to enable the use of unlabeled images, where thresholds are utilized to gradually [34] refine the model performance. Desai [35] proposed an active learning-based sampling strategy to select a set of highly representative labeled images for training and demonstrated the effectiveness of the proposed network on two publicly available satellite datasets. The BAS4Net model proposed by Sun et al. [36] uses the channel weighted multi-scale feature (CMF) module to balance the semantic and spatial information, and uses the boundary attention module (BAM) to weight the features with rich semantic boundary information to alleviate the boundary ambiguity. In the PiCoCo model [37], the semi-supervised segmentation of building footprint using pixel contrast and consistency learning has achieved considerable results. However, using semi-supervised learning for forest fire monitoring has rarely been reported. In this paper, we aim to bridge the gap, and address the problems (e.g., arbitrary shape and varying size of forest fire, and complex scene background) when applying semi-supervised leaning to forest fire monitoring.

### 2.2. Proposed Method

In this paper, we propose a semi-supervised learning network, SemiFSNet, for forest fire segmentation in UAV-acquired RGB imagery. Figure 1 shows the overall framework, which consists of supervised and unsupervised branches, where the encoder is shared for extracting the features from labeled and unlabeled inputs. First, an information deletion data enhancement strategy is performed on the labeled dataset by choosing different scales of grid to mask the images. This effectively alleviates the impact of the occlusions of fire caused by vegetation. For unlabeled data, a simple random cropping operation is applied for data augmentation. The augmented data including the labeled and unlabeled images are then input to the encoder network. To better extract the meaningful information from the forest fire region with varying shape and size, the encoder structure is constructed by replacing traditional convolution with dynamic convolution to enable the model to be automatically aware of the features at each scale of the forest fire image, thus facilitating the extraction of the multi-scale information. The encoder features are then refined by an attention module to suppress the interference caused by complex scene background.

Finally, the labeled data are fed to the classifier layer and the unlabeled data are non-linearly projected to the embedded feature space, where cross entropy (CE) loss and improved directional contrastive (DC) loss are used for training, respectively.

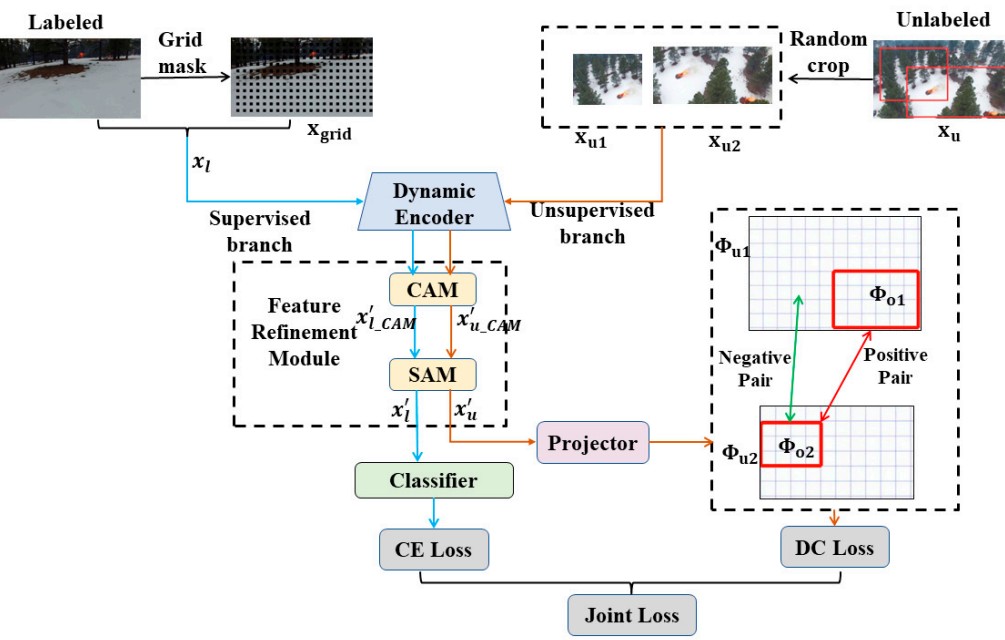

**Figure 1.** Overall structure of the proposed SemiFSNet.

### 2.2.1. Data Augmentation

Generally, the richer in information the labeled samples are, the stronger the generalization ability of the trained model. In our case, due to the limited number of labeled samples, it is essential to select an appropriate means to augment the data. Moreover, the potential occlusion of fire caused by vegetation needs to be considered. There are a few commonly used data augmentation techniques for remote sensing, such as CutOut [34], RErase [38] and HaS [39]. CutOut is performed by removing a square region of an image. RErase is used for the random deletion of a rectangle region in the image. HaS enhances the image by dividing it evenly into small chunks and then deleting them randomly. However, there is a significant risk that these data augmentation methods might remove parts of the forest fire (as illustrated in Figure 2), which degrades the model training. Inspired by the work in [40], we introduce Gridmask strategy for labeled data augmentation. The Gridmask strategy is one of the information-dropping-based data augmentation methods, which avoids excessive deletion while maintaining continuous regions.

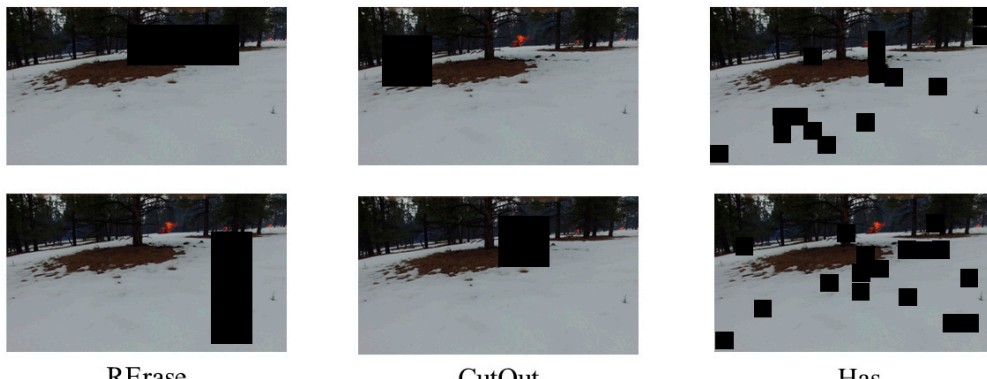

**Figure 2.** Images using different data augmentation techniques.

Specifically, Gridmask generates a grid-like mask of the same size as the original image. Gridmask has only two values, 0 and 1, where 0 denotes the black-masked area and 1 denotes the transparent area. The greater the number of 1s in mask, the greater the proportion of original information that will be retained. The smaller the number of 1s in the mask, the lower the proportion of information in the original image but the higher the risk of underfitting the model. There are, in total, four parameters (i.e., r, d, $\delta_x$, $\delta_y$) for the mesh (as shown in Figure 3) that affect the performance of extracted feature, where $\delta_x$ and $\delta_y$, respectively, denote the width and height of a dropped square with value 0, r denotes the retained proportion of the input image, and d denotes the length of a square.

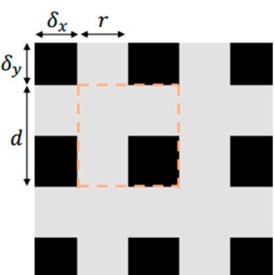

**Figure 3.** The dotted square denotes one unit of the mask.

Given the input images $X \in R^{H \times W \times C}$, $M \in \{0,1\}^{H \times W}$ denoting the binary mask, which stores pixels to be removed, the Gridmask data augmentation multiplies $X$ and $M$ to obtain the augmented image $Y$, i.e.,

$$Y = X \times M \tag{1}$$

According to [40], another parameter $k$ denotes the keep ratio of a mask $M$, which is defined as:

$$k = \frac{\text{sum}(M)}{H \times W}. \tag{2}$$

Ignoring incomplete units in the given mask, $k$ is related to r by:

$$k = 1 - (1-r)^2 = 2r - r^2. \tag{3}$$

The choice of d decides the size of a dropped square, where a small d helps to avoid failure cases. $\delta_x$ and $\delta_y$ control the shift of the mask. When parameters $r$ and d are given, the change of $\delta_x$ and $\delta_y$ enables the mask to cover all possible situations.

In the original Gridmask [40], when the value of d is between 96 to 224, the feature accuracy extracted by the model is higher, while the value of r is generally 0.3, 0.4 and 0.5. Considering that the fire point is small in the early stage of the fire, setting r to 0.5 may lead to the phenomenon that the small fire point is blocked, so we set r to 0.1, 0.3 and 0.4, respectively. In our case, to capture the occluded forest fire with varying size and shape, we design three different scales of Gridmask, where dropped squares with different scales are generated. We randomly select $\delta_x$ and $\delta_y$, ranging from 0 to $d-1$, and set (r, d) for different scales of masks as (120, 0.1), (100, 0.4) and (50, 0.3). The generated Gridmask maps are then multiplied with the original image to obtain the augmented sample sets. Figure 4 illustrates the use of Gridmask in data augmentation. For the same forest fire image, Gridmask maps with three scales (r, d) of (120, 0.1), (100, 0.4) and (50, 0.3) are used to cover it, and the occlusion parts and areas of different scales of Gridmask maps in the original image are different.

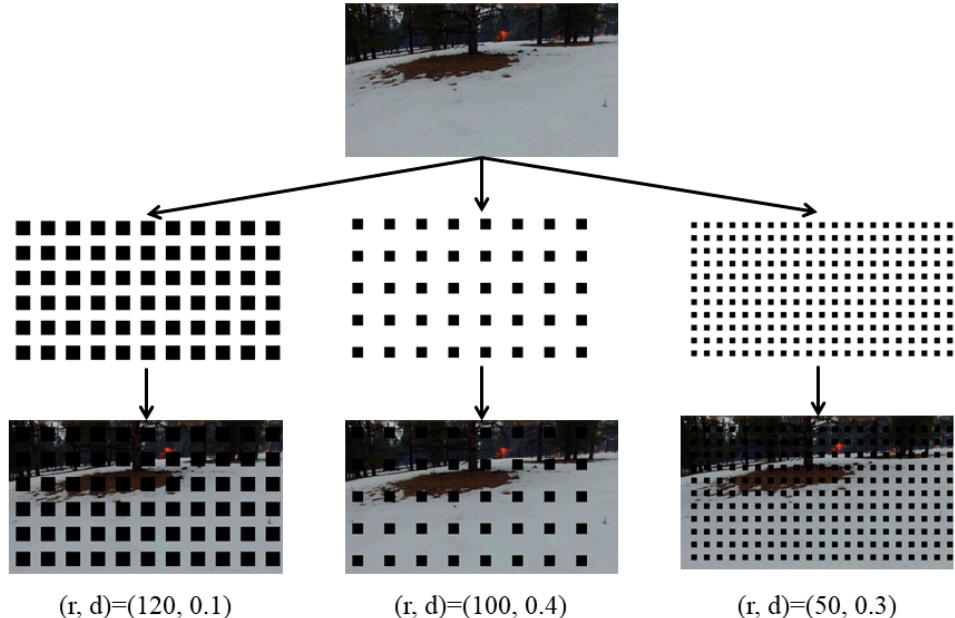

$(r, d)=(120, 0.1)$  $(r, d)=(100, 0.4)$  $(r, d)=(50, 0.3)$

**Figure 4.** Gridmask with different scales for data augmentation.

### 2.2.2. Dynamic Encoder

Traditional convolution tends to use a single convolution kernel per layer with the kernel size fixed, leading to limited representation ability. In this paper, to increase the capability in representing forest fire, we propose the dynamic encoder module for replacing the typical fixed-size convolution with dynamic convolution, where a set of L parallel convolution kernels are introduced [41]. Dynamic convolution computes multiple attention weights for each kernel and aggregates all L kernels based on the computed weights, where the weights are calculated based on the attention operation. The size of the dynamic convolution kernel varies with the size of the feature map, and this property makes dynamic convolution suitable for the task of identifying the target with varying scales, i.e., forest fire.

Specifically, given an input $x$ and multiple (L) linear functions $\left\{ W'^T_L x + b'_L \right\}$, the squeeze-and-excitation attention [42] is applied to compute the kernel attention $\{\pi_L(x)\}$. Unlike the work in [42], the attentions are computed over convolution kernels instead of over output channels. After obtaining $\{\pi_L(x)\}$, the aggregated weight $W'(x)$ and bias $b'(x)$ are computed based on attention weighed as follows:

$$W'(x) = W'_1 * \pi_1(x) + W'_2 * \pi_2(x) + \cdots + W'_l * \pi_k(x) = \sum_{l=1}^{k} (\pi_l(x)W'_l), \qquad (4)$$

$$b'(x) = b'_1 * \pi_1 + b'_2 * \pi_2 + \cdots + b'_k * \pi_k = \sum_{l=1}^{k} (\pi_l(x)b'_l), \qquad (5)$$

where $W'_l$ and $b'_l$, respectively, represent the weight and bias of the dynamic perceptron, and $\pi_l(x)$ denotes the attention weight of the $l$th linear function $\left\{ W'^T_l x + b'_l \right\}$. The constraints of $\pi_l(x)$ are

$$0 \leq \pi_k(x) \leq 1, \sum_{l=1}^{k} \pi_l(x) = 1. \qquad (6)$$

The output of dynamic convolution $y$ is obtained by:

$$y = g\left( W'^T(x)x + b'(x) \right). \qquad (7)$$

where $g$ denotes the activation function, and RELU activation function is used in $g$ in the experiment.

This is followed by batch normalization and an activation function (ReLU in our case). Dynamical convolution automatically assembles the different kernels, increasing the model capability to extract useful information. A dynamic convolution layer is shown in Figure 5.

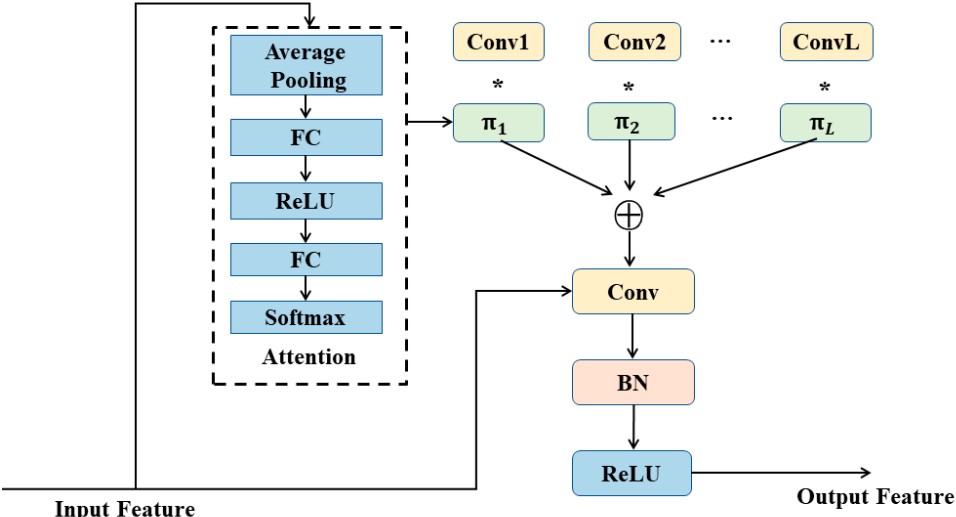

**Figure 5.** Dynamic convolution structure diagram.

In our encoder network, the $3 \times 3$ convolutions are replaced with dynamic convolutions over every block of the backbone, i.e., Resnet50, as shown in Figure 6. By using the dynamic convolution kernel, the dynamic encoder $f_{DE}$ is generated and the output features $F_{DE}$ are defined as:

$$F_{DE} = f_{DE}(Mix(x_l + x_u), \tag{8}$$

where $Mix()$ denotes the batches comprising labeled images $x_l$ and unlabeled images $x_u$.

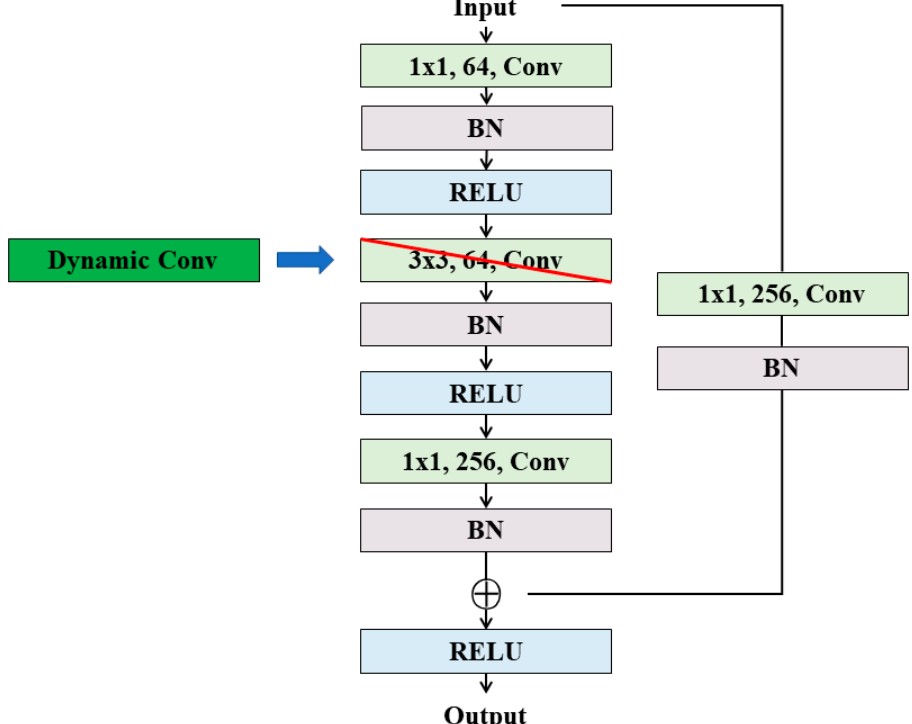

**Figure 6.** Replacing a $3 \times 3$ convolution with a dynamic convolution in Resnet50.

### 2.2.3. Feature Refinement Module

Forest fire segmentation using remote sensing imagery tend to be more difficult compared with other common images due to the more complex scene background. The attention mechanism is essential to enable the model to focus more on the meaningful information and ignore irrelevant information [43]. Thus, we propose a feature refinement module by integrating an attention mechanism to the output features of the encoder, which aids the model to highlight the useful fire features while reducing the interference of background information. According to [44], to obtain the more discriminative and powerful feature, the spatial and channel-wise attention is incorporated in a CNN. To this end, we propose to integrate the channel attention module (CAM) with the spatial attention module (SAM) in a concatenating way (as shown in Figure 7), where the output features are processed along two independent dimensions and thus resulting in a more discriminative representation.

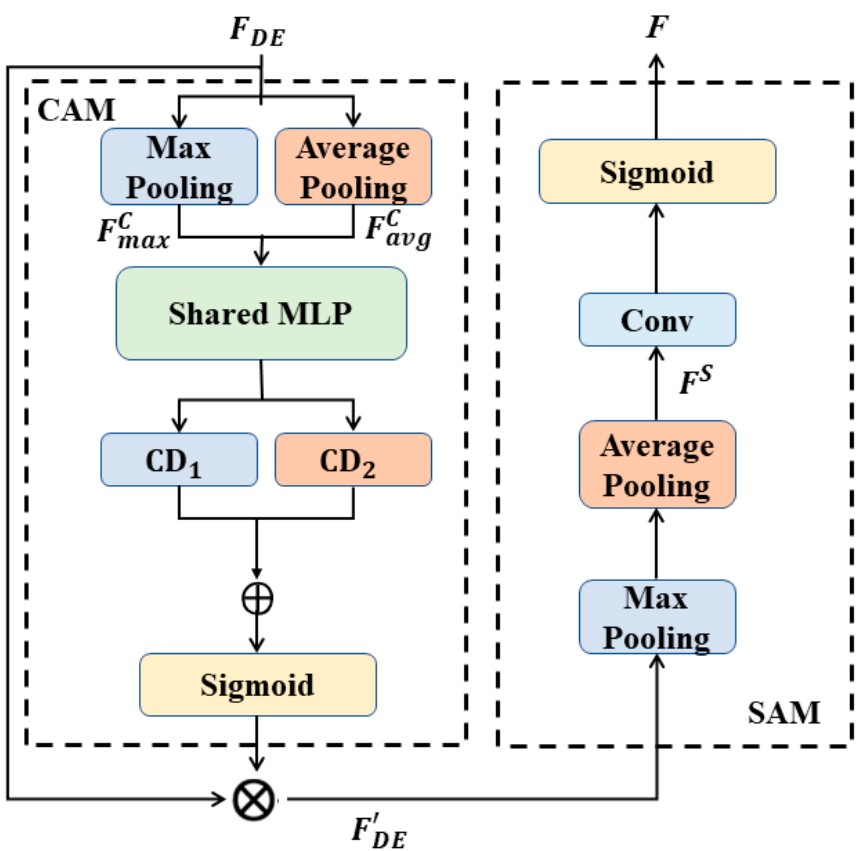

**Figure 7.** Feature refinement module with CAM and SAM.

Specifically, the output feature $F_{DE}$ of the dynamic encoder are first aggregated using max pooling and average pooling to generate spatial descriptors $F_{max}^{C}$ and $F_{avg}^{C}$. The two descriptors are then fed to a shared multi-layer perceptron (MLP) network and merged using element-wise summation. The channel attention is computed by:

$$CA(x) = \sigma(MLP(AvgPool(F_{DE})) + MLP(MaxPool(F_{DE}))), \qquad (9)$$

where $\sigma$ denotes the sigmoid function, and $W_1$ and $W_0$ are the *MLP*. The generated channel attention is multiplied by $F_{DE}$ to obtain the CAM feature maps $F'_{DE}$, i.e.,

$$F'_{DE} = CA(F_{DE}) \otimes F_{DE} \qquad (10)$$

After obtaining $F'_{DE}$, the spatial attention is computed by applying max pooling and average pooling, resulting in a descriptor $F^S$. Such pooling operations highlight

the informative forest fire regions. The output is then fed to a $7 \times 7$ convolution layer followed by a sigmoid function to generate spatial attention maps. The spatial attention is computed by:

$$SA(F'_{DE}) = \sigma\left(\text{Conv}\left(F^S\right)\right). \tag{11}$$

The feature maps and $F'_{DE}$ are element-wise multiplied to obtain the SAM feature maps, i.e., the final refined features are computed by:

$$F = SA\left(F'_{DE}\right) \otimes F'_{DE} \tag{12}$$

### 2.2.4. Consistency Regularization for Unsupervised Learning

There are two batches of inputs in our semi-supervised segmentation framework, i.e., $x_l$ and $x_u$, denoting labeled and unlabeled images, respectively. The dynamic encoder architecture with the feature refinement module $f_{DE+R}$ embeds the labeled images in the feature maps $F_l = f_{DE+R}(x_l)$, and the decoder (classifier) makes predictions $p_l = f_C(F_l)$. The learning process is provided by ground-truth labels $y_l$ using the standard cross entropy loss $\mathcal{L}_{ce}$. For the unlabeled images, we randomly crop the two patches $x_{u1}$ and $x_{u2}$ with an overlapping region $x_o$, and then augment $x_{u1}$ and $x_{u2}$ using low-level augmentation. The two augmented patches are then fed to the encoder model $f_{DE+R}$ to obtain the feature map $F_{u1}$ and $F_{u2}$. Following the work in [27], the two features are embedded using non-linear projection as $\Phi$, i.e.,

$$\phi_{u1} = \Phi(F_{u1}), \tag{13}$$

$$\phi_{u2} = \Phi(F_{u2}), \tag{14}$$

As illustrated in Figure 1. The features from the overlapping areas in $\phi_{u1}$ and $\phi_{u2}$ are, respectively, referred to as $\phi_{o1}$ and $\phi_{o2}$, where the $\phi_{o1}$ and $\phi_{o2}$ should remain consistent under different contexts.

A context-ware consistency constraint, i.e., DC loss [27], is used to enable the features from the overlapping areas to remain consistent with each other. The DC loss is inspired by the contrastive loss, which pulls the positive samples closer while separating the negative samples belonging to other classes. In our case, the features from overlapping locations $\phi_{u1}$ and $\phi_{u2}$ are regarded as a positive pair, as they share the same pixels despite being under different contexts, and any two features in $\phi_{u1}$ and $\phi_{u2}$ from different locations are regarded as a negative pair. Unlike contrastive loss, the DC loss further exploits a directional alignment for the positive pairs, which effectively prevents the high confident feature from suppressing the low confident one. This is necessary, as the prediction with higher confidence tends to be more accurate, and features with lower confidence need to be aligned to their counterpart with higher confident. The confidence of each feature $\phi_{u1}$ is measured using maximum probability among all classes, i.e., $\max(\mathcal{C}(f_i))$. For the t-th unlabeled image, the DC loss $\mathcal{L}_{dc}^t$ is computed as:

$$l_{dc}^t(\phi_{o1}, \phi_{o2}) = -\frac{1}{N} \sum_{h,w} \mathcal{M}_d^{h,w} \cdot \log \frac{r\left(\phi_{o1}^{h,w}, \phi_{o2}^{h,w}\right)}{r\left(\phi_{o1}^{h,w}, \phi_{o2}^{h,w}\right) + \sum_{\phi_n \in \mathcal{F}_u} r\left(\phi_{o1}^{h,w}, \phi_n\right)} = 1\left\{ max\mathcal{C}\left(f_{o1}^{h,w}\right) < max\mathcal{C}\left(f_{o2}^{h,w}\right)\right\}, \tag{15}$$

$$\mathcal{L}_{dc}^t = l_{dc}^t(\phi_{o1}, \phi_{o2}) + l_{dc}^t(\phi_{o2}, \phi_{o1}), \tag{16}$$

where $N$ is the number of spatial locations of overlapping area, h and w represent the 2-D spatial locations, $\phi_n$ denotes the negative counterpart of feature $\phi_{o1}^{h,w}$ and $r$ represents the exponential function of the cosine similarity s between two features with a temperature $\tau$, i.e., $r(\phi_1, \phi_2) = \exp(\frac{s(\phi_1, \phi_2)}{\tau})$, and $\mathcal{F}_u$ denotes the set of negative samples. Since more negative samples result in better performance, a memory bank is used to store the features

from the last few batches to acquire more negative samples [27]. The final loss is then computed by summing the loss of each image, i.e.,

$$\mathcal{L}_{dc} = \frac{1}{T} \sum_{t=1}^{T} \mathcal{L}_{dc}^{t}, \tag{17}$$

where $T$ denotes the number of unlabeled images within a training batch.

### 2.2.5. Joint Loss Function

The joint loss function of the proposed semi-supervised-based method comprises two parts: cross entropy loss $\mathcal{L}_{ce}$ for supervised learning and consistency constraint loss $\mathcal{L}_{dc}$ for unsupervised learning. It is defined as:

$$\mathcal{L} = w_l \mathcal{L}_{ce} + w_{un} \mathcal{L}_{dc} \tag{18}$$

where $w_l$ and $w_{un}$, respectively, denote the supervised loss weight and unsupervised loss weight, which balance the contributions of two loss parts. In our experiments, numerous experiments have shown that $w_l$ and $w_{un}$ are set to 0.7 and 0.4, respectively.

## 3. Results

To verify the effectiveness of the proposed method SemiFSNet for forest fire segmentation in UAV images, we conducted intensive experiments on two publicly available datasets, the Flame dataset [45] and Corsican dataset [46].

### 3.1. Datasets

The Flame dataset [45] is created by researchers from Northern Arizona University, where the samples are collected using UAVs during the prescribed burning of piles of debris in an Arizona pine forest. The dataset includes video recordings and heat maps taken by infrared cameras. The captured videos and images are labeled by frame and aid researchers to easily build models. The dataset contains a total of 2003 forest fire images, each of which is $3840 \times 2160$ in size. The fire points are generally freshly started or extinguished, with the fire points taking up a relatively small proportion of the whole image.

The Corsican dataset [46] is created by the Environmental Sciences Laboratory of the University of Corsica, which includes 1136 forest fire images, each with the size of $1024 \times 729$ captured using an RGB camera. The background of the images varies considerably, making the segmentation difficult. To evaluate the performance in segmenting remote small fire, we selected 654 images with smaller fire spots for our experiment. Note that it is more challenging to segment small-sized fire due to its low resolution and less spatial information. Figure 8 shows some samples from the Flame dataset and Corsican dataset.

### 3.2. Implementation Details

DeepLabV3+ is employed as the encoder-decoder network of our proposed method due to its effectiveness in extracting multi-scale information, where ResNet50 is used as the backbone. The labeled and unlabeled images are first resized to $224 \times 224$, and then augmented using Gridmask and random crop. During model training, SGD optimizer is used and the learning rate, weight decay and momentum are set to 0.01, 0.0001 and 0.9, respectively. The values are gradually decreased to zero following the polynomial decay schedule. The training batch size is set to 4 and the model is trained for 100 epochs. In total, 20% of the samples from two datasets are randomly selected as the testing set, and the remaining samples as the training set. All experiments were performed on an NVIDIA RTX3090.

Flame Dataset                    Corsican Dataset

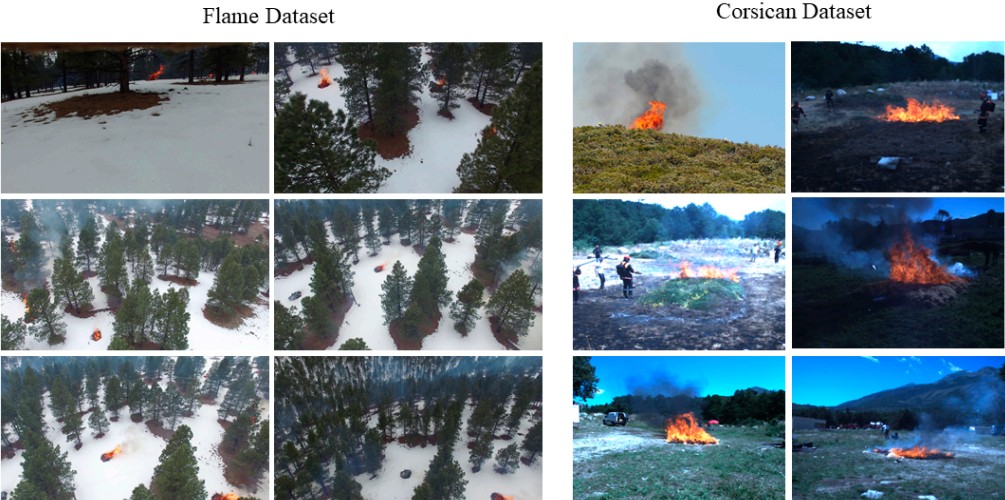

**Figure 8.** Selected samples from Flame dataset and Corsican dataset.

The intersection-over-union (*IoU*) for each class is employed as the evaluation metric. IoU is also known as the Jaccard Index, and is a statistic indicating the similarity and diversity of samples. In semantic segmentation, *IoU* denotes the ratio of the intersection of the pixel-wise classification results and ground truth, and is used to determine the spatial overlap between the prediction and ground truth, i.e.,

$$IoU = \frac{TP}{TP + FP + FN},$$ (19)

where *TP* denotes true positives, *FP* denotes the false positives, *TN* denotes the true negatives and *FN* denotes false negatives. *TP* refers to positive samples predicted by the model as positive class, *FP* refers to negative samples predicted by the model as positive class, *TN* refers to negative samples predicted by the model as negative class, and *TN* refers to positive samples predicted by the model as negative class.

In multiclass semantic segmentation, mean *IoU* (mIoU) is mainly used as an evaluation metric. There are only two class categories in our case, i.e., forest fire and background. Since the proportion of fire area is small, and the scene background occupies the large portion of an entire image, if mIoU is used as the evaluation metric then the *IoU* of the background has a significant influence on the overall mIoU, which have no value. Therefore, this paper only uses the *IoU* of forest fire as our evaluation indicator.

### 3.3. Comparison with the State-of-the-Art Methods

Flame dataset. To assess the effectiveness of the proposed method, we made comparisons with state-of-the-art methods, including Adv-Semi [24], CycleGAN [25], ST++ [26], CAC [27], CCT [47], and ECS [48]. We implemented these methods within a unified framework following their official code, where the same base backbone, i.e., ResNet, is used and the same data lists are used for training and testing. We compared the proposed method under the setting with various labeled data proportions of the training set, i.e., 2/8, 3/7, 5/5 and full labeled data. In the full data setting, images fed to the unsupervised branch are simply collected from the labeled set. We ensured that all other parameters are consistent during the experiment. The experimental results on the Flame dataset are shown in Table 1.

Table 1 shows that the segmentation results using the proposed method on the Flame dataset consistently outperforms the state-of-the-art methods on all data pro-portions. This shows the superiority of the proposed method in identifying small objects. Under 2/8 data proportion, the IoU performance of the proposed method is 11.5% higher than the Adv-Semi model and 3.8% higher than the ST++. These demonstrate that the proposed method effectively addresses the situation with limited labeled data. Compared with the

SupOnly (i.e., using only supervised loss), the proposed method using the supervised loss and unsupervised loss achieves a constant increase in terms of IoU under all data proportions. The idea of self-correction has been exploited in CCT and ECS by creating the Correction Network and Flaw Detector, respectively, to amend the defects in predictions. Our method not only uses self-correction, but also optimizes the entire model in the feature extraction stage to strengthen the extraction of small-target fire features. As can be seen in Table 1, our method outperforms CCT by 3.1%, 1.3% and 0.6%, and outperforms ECS by 4.1%, 2.5% and 2.5%, under three different proportions.

**Table 1.** Comparison with the baseline (SupOnly, i.e., using only supervised loss) and other state-of-the-art methods on Flame dataset with 2/8, 3/7, 5/5 and full labeled data.

| Networks | The Ratio of Labeled and Unlabeled for Training | | | |
|---|---|---|---|---|
| | 2/8 | 3/7 | 5/5 | Full |
| CAC | 62.5% | 65.4% | 67.7% | 69.8% |
| ST++ | 60.6% | 67.6% | 67.9% | N/A |
| Adv-semi | 52.9% | 52.6% | 49.7% | N/A |
| CycleGAN | 49.7% | 55.4% | 63.6% | N/A |
| CCT | 61.3% | 64.1% | 68.4% | N/A |
| ECS | 60.3% | 62.8% | 66.5% | N/A |
| Ours | 64.4% | 65.4% | 69.0% | N/A |

To visually illustrate the segmentation performance of the proposed method, we also present a visual comparison under the 2/8 proportion with the state-of-the-art methods, as shown in Figure 9.

As the target forest fire in the Flame dataset is small, we provide the enlarged comparison for clearer illustration in Figure 10. As shown in Figures 9 and 10, the proposed method accurately localizes and segments the small fire spots, and the segmentation results are almost consistent with the Ground truth mask both in terms of size and boundary. The prediction results using ST++, Adv-Semi and CycleGAN have relatively large errors, and their predicted fire points are smaller than the Ground truth. The CAC predictions are closer to the Ground truth, but there are more dispersed irrelevant points. The results of CCT and ECS are close to the ground truth, but the segmentation results of our method are more refined, and the contour features are more obvious. These points are caused by the similarity between smaller fire and dead twigs or bare soil. Overall, the experiment results show that SemiFSNet improves the accuracy for the segmentation of forest fire, especially with small-sized fire.

Corsican dataset. Compared with the Flame dataset, the images in the Corsican dataset have relatively larger fire targets. We use this dataset to further evaluate the effectiveness of SemiFSNet on aerial images with large fire points by comparing its performance with state-of-the-art methods quantitatively and qualitatively. The results are shown in Table 2 and Figure 11. Once again, the proposed method consistently outperforms the other methods under all data proportion except for CycleGAN. The performance of CycleGAN is higher than the proposed method by 1.6% when the ratio of labeled and unlabeled is 2/8. However, CycleGAN training was unstable, and IoU fluctuated greatly under different proportions. In addition, the IoU of the proposed method under a different semi-supervised learning setting (i.e., 80.7%, 75.2% and 80.3%) is even far better than using a full supervised setting (i.e., 65.4%). This shows that replacing labeled images with unlabeled images has significant benefits for model learning, alleviating the negative impact of noise caused by manual labeling on model training. At the same time, the requirement of manual annotation is reduced. Figure 11 visually shows that the results using the proposed method outperform those of other methods in terms of the boundary and shape of the forest fire.

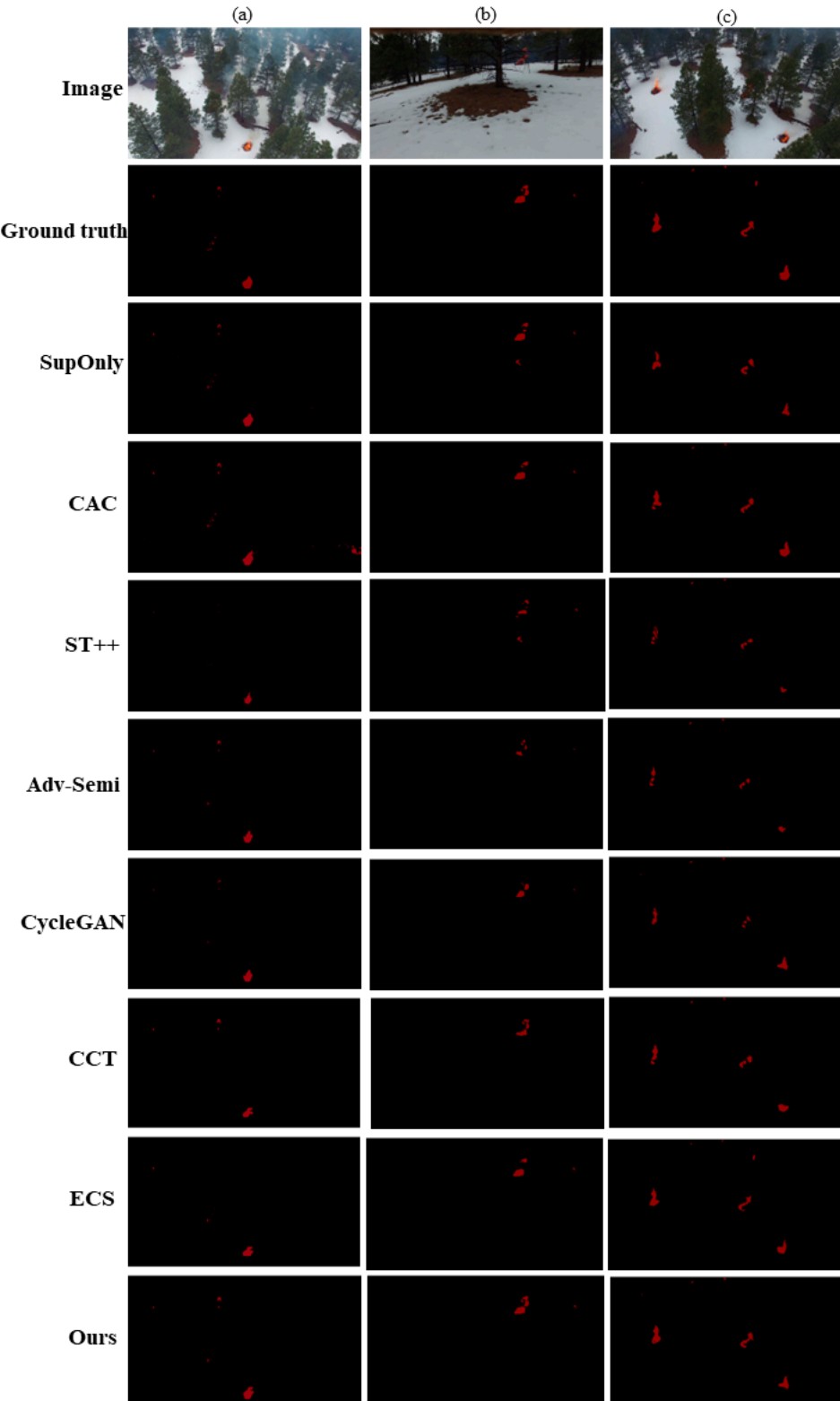

**Figure 9.** Visualization of segmentation results using the proposed method and state-of-art methods on the Flame dataset, where subfigure (**a**) shows the image of a single fire point taken by UAV from high altitude, subfigure (**b**) shows the image of fire point taken by UAV from low altitude, and subfigure (**c**) shows the image of multiple fire points taken by UAV.

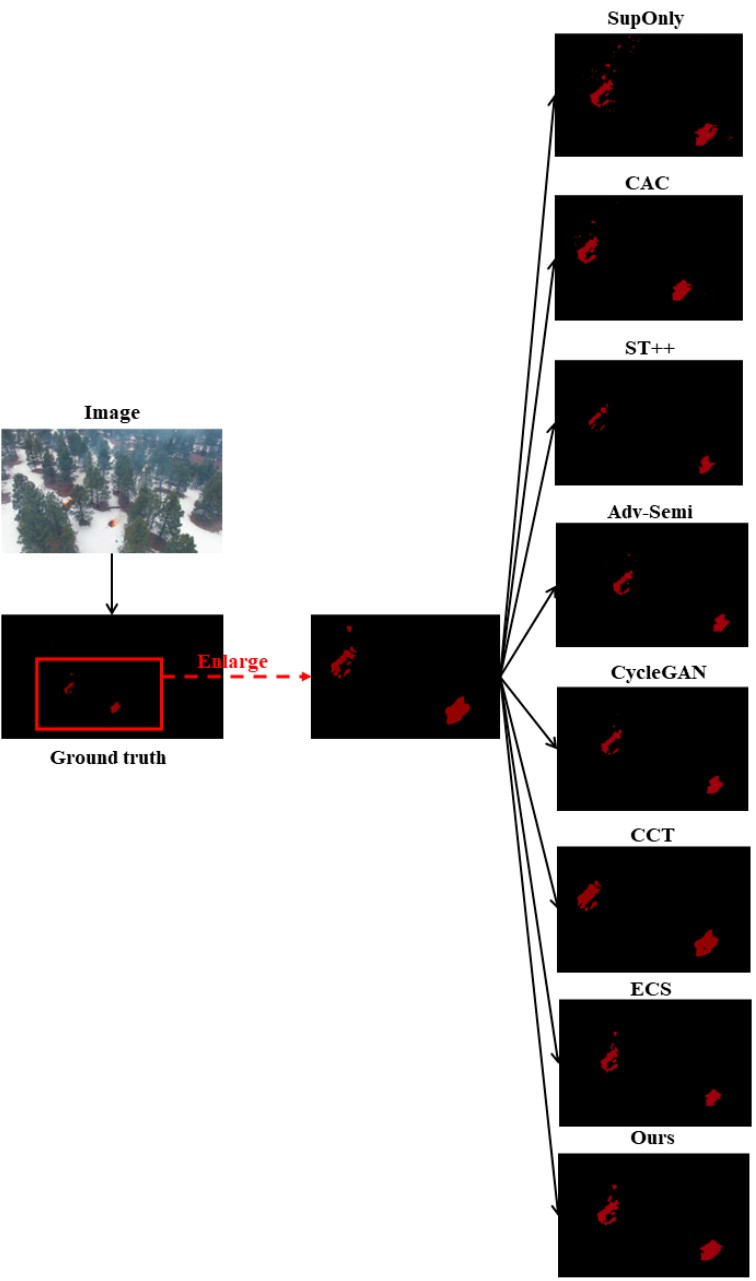

**Figure 10.** Enlarged visualization of segmentation results using the proposed method and state-of-art methods on the Flame dataset.

**Table 2.** Comparison with the baseline and other state-of-the-art methods on Corsican dataset with 2/8, 3/7, 5/5 and full labeled data.

| Networks | The Ratio of Labeled and Unlabeled for Training | | | |
|---|---|---|---|---|
| | **2/8** | **3/7** | **5/5** | **Full** |
| CAC | 70.5% | 75.9% | 76.2% | 65.4% |
| ST++ | 63.1% | 78.6% | 77.1% | N/A |
| Adv-semi | 73.9% | 76.9% | 80.1% | N/A |
| CycleGAN | 69.7% | 73.9% | 80.0% | N/A |
| CCT | 68.3% | 70.4% | 75.3% | N/A |
| ECS | 70.2% | 72.7% | 75.4% | N/A |
| Ours | 80.7% | 75.2% | 80.3% | N/A |

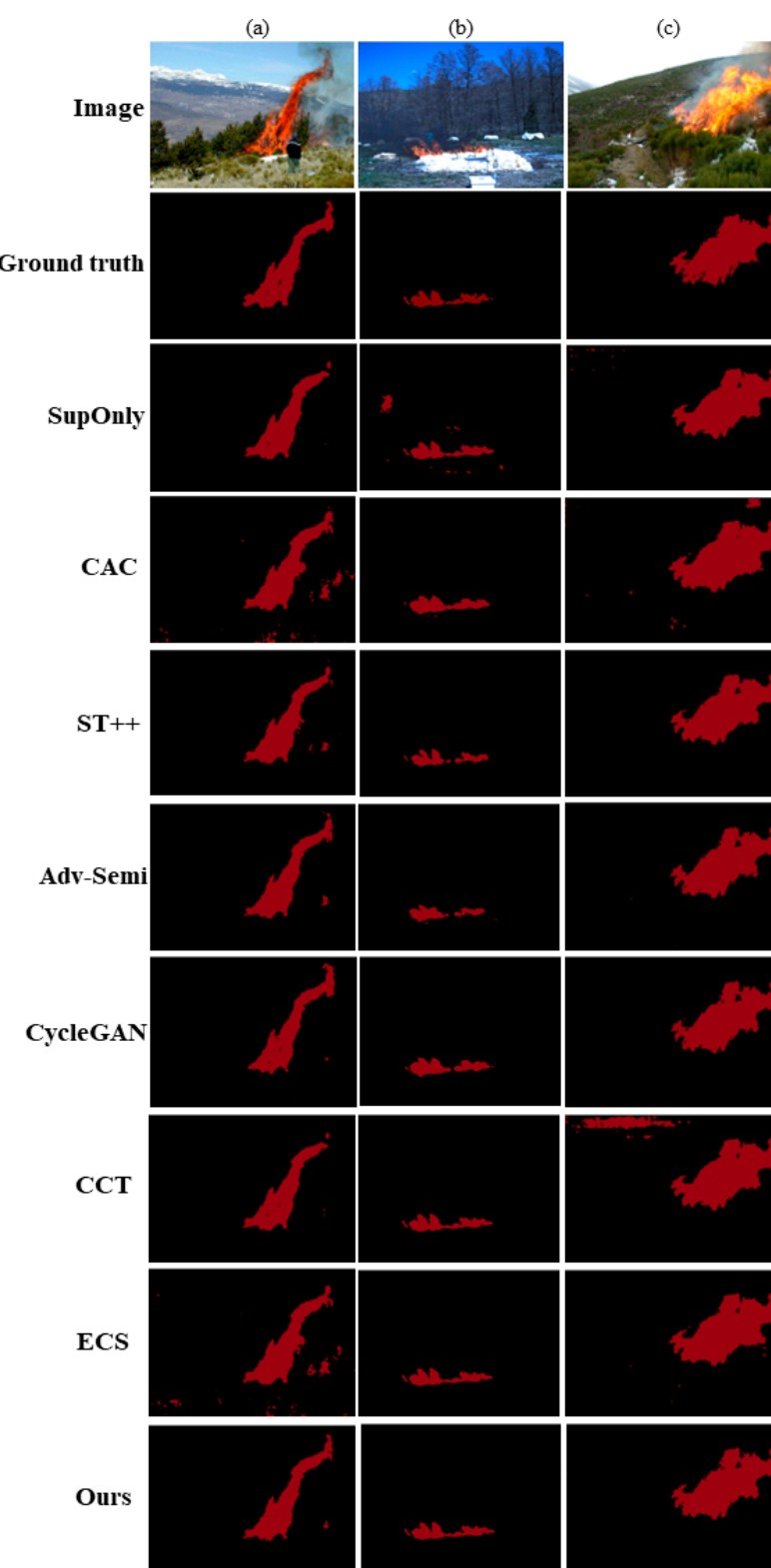

**Figure 11.** Visualization of segmentation results using the proposed method and state-of-art methods on the Corsican dataset, where (**a**–**c**) shows the samples with different background.

### 3.4. Ablation Study

To better demonstrate the efficacy of each component of SemiFSNet, an ablation study was performed on the same two datasets. There are three key components in SemiFSNet, i.e., Gridmask, Dynamic Encoder with Dynamic Convolution, and the attention-based

feature refinement module. Intensive experiments were conducted using the baseline method [27] with the individual components and their combinations under different data proportions. The results are shown in Tables 3 and 4.

**Table 3.** Results of ablation study on the Flame dataset. (✔ denotes the module is included in the method.)

| Allocation Strategy | Baseline | Gridmask | Dynamic-Convolution | CAM + SAM | IOU |
|---|---|---|---|---|---|
| | ✔ | | | | 60.3% |
| | ✔ | ✔ | | | 61.1% |
| | ✔ | | ✔ | | 62.5% |
| 2/8 | ✔ | | | ✔ | 63.1% |
| | ✔ | ✔ | ✔ | | 63.3% |
| | ✔ | ✔ | | ✔ | 63.2% |
| | ✔ | | ✔ | ✔ | 64.2% |
| | ✔ | ✔ | ✔ | ✔ | 64.4% |
| | ✔ | | | | 61.5% |
| | ✔ | ✔ | | | 62.5% |
| | ✔ | | ✔ | | 61.9% |
| 3/7 | ✔ | | | ✔ | 62.3% |
| | ✔ | ✔ | ✔ | | 63.7% |
| | ✔ | ✔ | | ✔ | 64.2% |
| | ✔ | | ✔ | ✔ | 65.2% |
| | ✔ | ✔ | ✔ | ✔ | 65.4% |
| | ✔ | | | | 64.7% |
| | ✔ | ✔ | | | 64.9% |
| | ✔ | | ✔ | | 65.6% |
| 5/5 | ✔ | | | ✔ | 67.3% |
| | ✔ | ✔ | ✔ | | 66.4% |
| | ✔ | ✔ | | ✔ | 67.3% |
| | ✔ | | ✔ | ✔ | 68.7% |
| | ✔ | ✔ | ✔ | ✔ | 69.0% |

**Table 4.** Results of ablation study on the Corsican dataset. (✔ denotes the module is included in the method.)

| Allocation Strategy | Baseline | Gridmask | Dynamic-Convolution | CAM + SAM | IOU |
|---|---|---|---|---|---|
| | ✔ | | | | 78.0% |
| | ✔ | ✔ | | | 78.6% |
| | ✔ | | ✔ | | 79.2% |
| 2/8 | ✔ | | | ✔ | 79.4% |
| | ✔ | ✔ | ✔ | | 79.3% |
| | ✔ | ✔ | | ✔ | 79.4% |
| | ✔ | | ✔ | ✔ | 80.1% |
| | ✔ | ✔ | ✔ | ✔ | 80.7% |
| | ✔ | | | | 73.2% |
| | ✔ | ✔ | | | 74.0% |
| | ✔ | | ✔ | | 73.8% |
| 3/7 | ✔ | | | ✔ | 73.6% |
| | ✔ | ✔ | ✔ | | 74.2% |
| | ✔ | ✔ | | ✔ | 74.7% |
| | ✔ | | ✔ | ✔ | 74.8% |
| | ✔ | ✔ | ✔ | ✔ | 75.2% |
| | ✔ | | | | 77.9% |
| | ✔ | ✔ | | | 78.2% |
| | ✔ | | ✔ | | 78.8% |
| 5/5 | ✔ | | | ✔ | 78.0% |
| | ✔ | ✔ | ✔ | | 79.2% |
| | ✔ | ✔ | | ✔ | 79.6% |
| | ✔ | | ✔ | ✔ | 80.0% |
| | ✔ | ✔ | ✔ | ✔ | 80.3% |

As can be seen from Tables 3 and 4, the labeled dataset is expanded by using Gridmask data augmentation so that the model achieves better segmentation results with a small amount of labeled data. Dynamic convolution leads to the increased performance on both datasets, where the features of fire with small or multiple scales are enhanced. Attention-based feature refinement significantly improves the IoU by focusing on the salient information, i.e., fire while suppressing the complex scene background. By combining the baseline with all three components, our method improves the segmentation of forest fire on the Flame dataset by up to 4.1% (for 2/8 data proportion). This demonstrates the

efficacy of our method on the segmentation of small fire with a limited number of samples. Similarly, for the Corsican dataset, our SemiFSNet yields a much higher performance than the method without Gridmask, Dynamic Convolution and Feature Refinement for all data proportions. Overall, SemiFSNet using Gridmask, Dynamic Convolution and Feature Refinement considerably improves the performance of forest fire segmentation in aerial images, where a limited number of labeled samples are used for model training, thus significantly reducing the labor cost.

## 4. Discussion

To better show the performance of the three key components on different numbers of labeled data, we generated the line graphs in Figure 12. The figure shows that SemiFSNet integrates all three components to consistently outperform the other methods without or with the use of individual component, regardless of the number of labeled data.

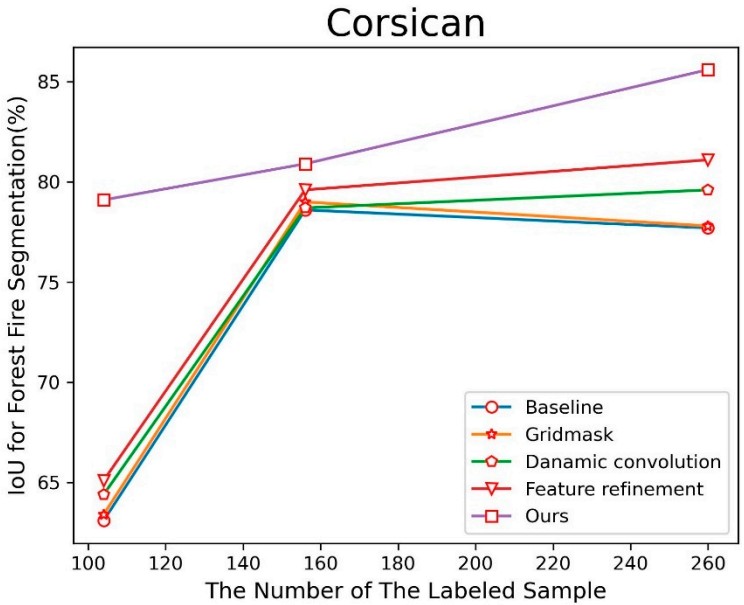

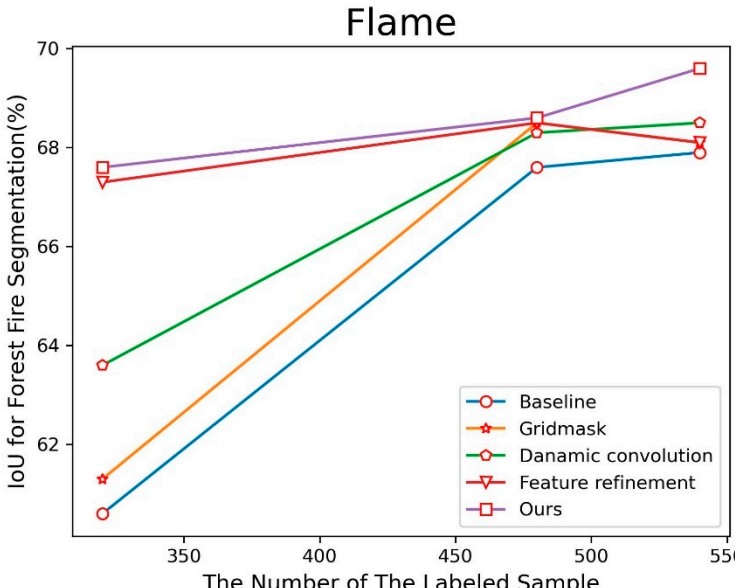

**Figure 12.** IoU for forest fire segmentation under various numbers of labeled images.

The loss reflects the error between the actual and predicted values, and the loss trend reflects the state of the model learning. It is expected that both the training loss and the validation (testing) loss will gradually decrease and converge, indicating the good feature learning by the model. The trend of the training loss and the validation loss can be used to determine whether the model is overfitting or underfitting [49]. The training loss of a general semi-supervised model comprises supervised and unsupervised, i.e., supervised loss and unsupervised loss. We visualize the supervised loss, unsupervised loss and validation loss of our model under different semi-supervised allocation strategies and determine whether there are any anomalies during training and testing. Figures 13 and 14 show the visualization of the loss under all allocation strategies on the Flame dataset and Corsican dataset, respectively.

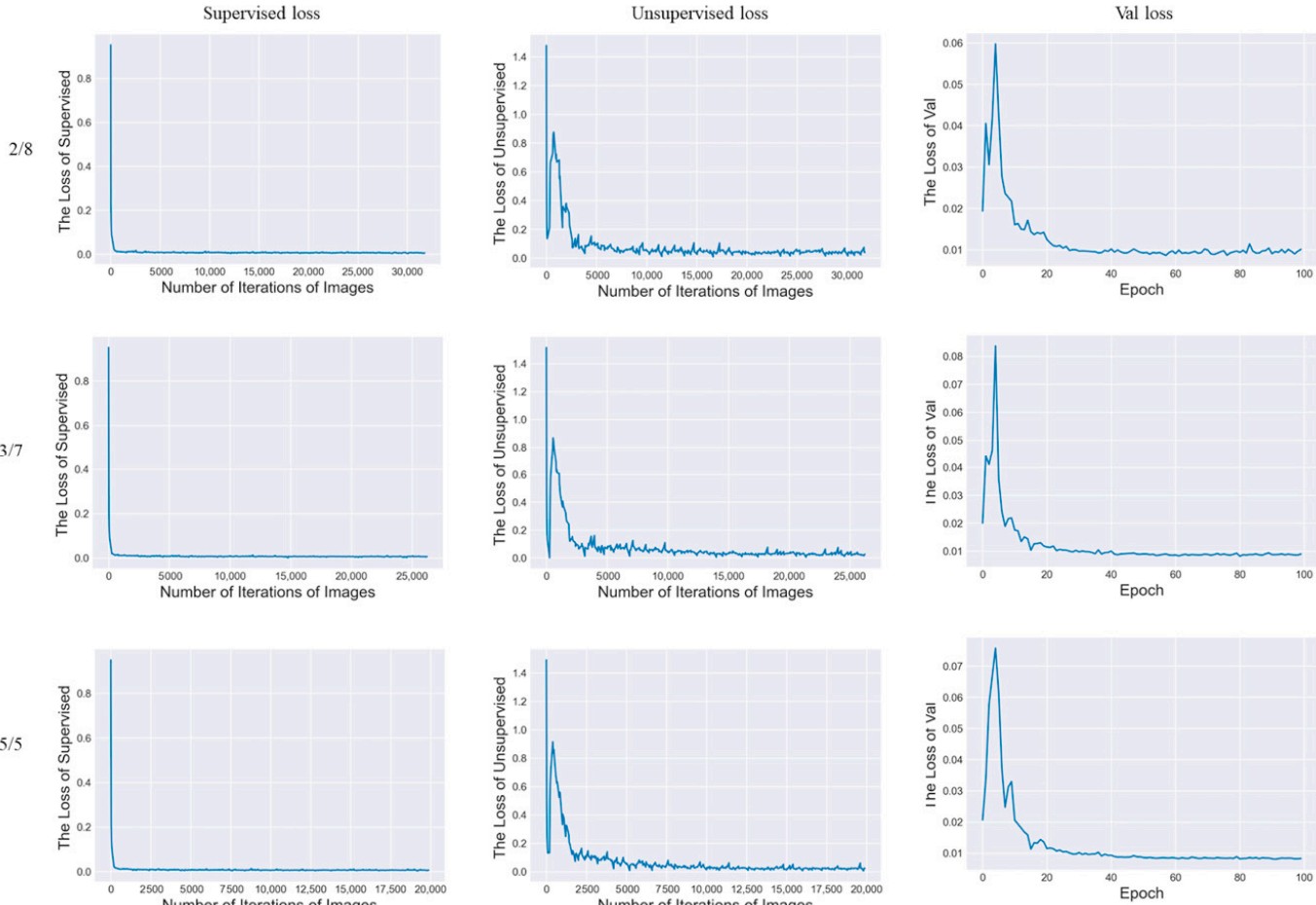

**Figure 13.** Loss curve for supervise training loss, unsupervised training loss and validation loss on the Flame dataset.

Both Figures 13 and 14 show that the supervised loss, unsupervised loss and validation loss display a trend of decreasing and then converging during the model training. This indicates that our model keeps adjusting the direction of convergence, making the model adjust to the best state. Judging from the convergence of the loss, the model is trustworthy and reliable.

We also compared the total parameters of our model with those of other models, and the total parameters of each model are shown in Table 5. The number of parameters of our model is 2.22 M less than that of CycleGAN's model. Considering the stability and number of parameters of our model in the segmentation of forest fire datasets at various proportions, our model is significantly better than CycleGAN's model. The number of parameters in our model is only 0.1 M higher than that in the CAC model and 0.2 M higher

than that in the ST++ model. The whole experimental results show that the performance of our model in forest fire image segmentation is far better than those two models.

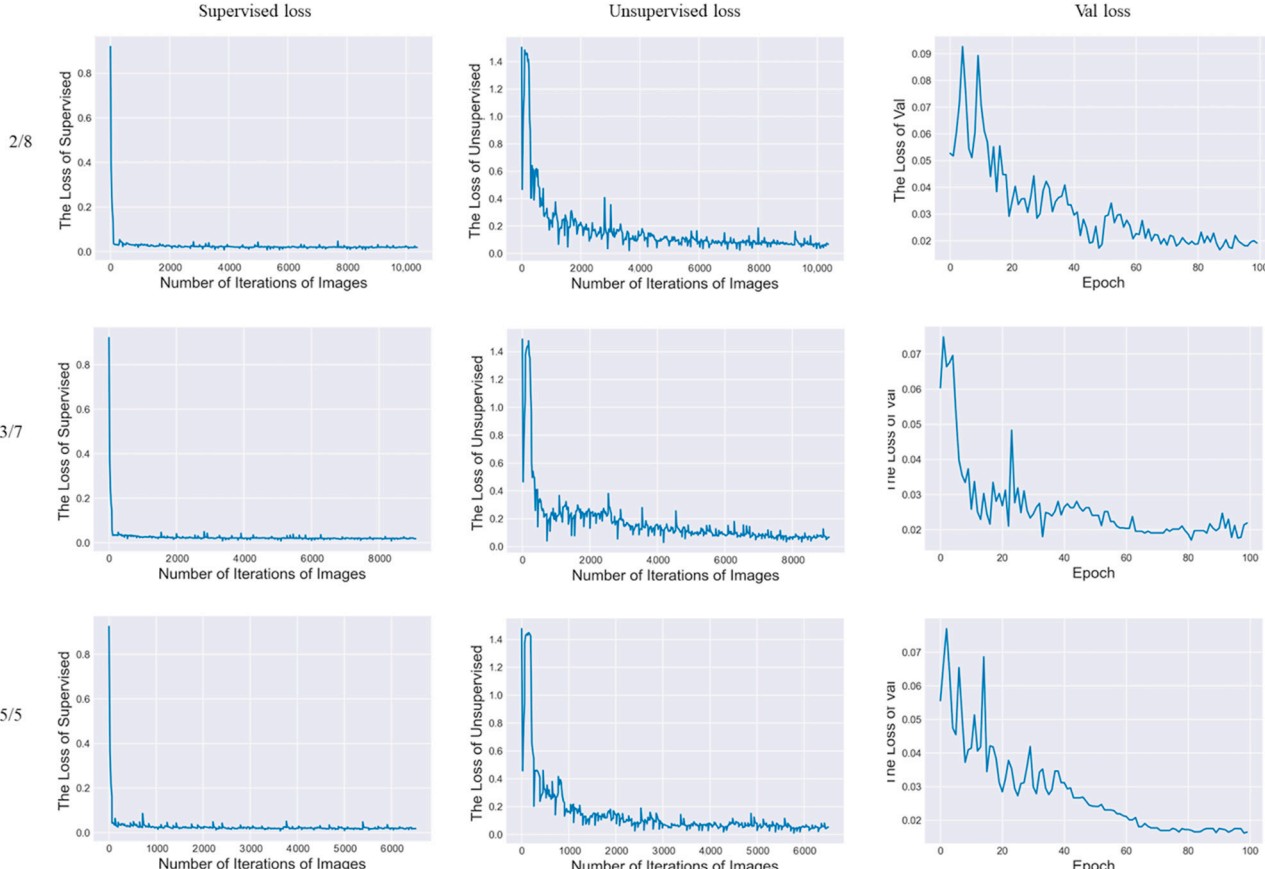

**Figure 14.** Loss curve for supervise training loss, unsupervised training loss and validation loss on the Corsican dataset.

**Table 5.** Parameters of each model.

|  | **CAC** | **ST++** | **Adv-Semi** | **Cycle-GAN** | **Ours** |
|---|---|---|---|---|---|
| Parameters | 40.4 M | 40.3 M | 44.05 M | 42.72 M | 40.5 M |

## 5. Conclusions

This paper focuses on solving the problem of forest fire segmentation in UAV-acquired RGB images, where semi-supervised learning techniques are applied to save human resources. We propose an effective SemiFSNet, which aims to address the challenges faced by existing remote sensing fire segmentation where the fire could: (1) be partly occluded by vegetation; (2) have varying size and shape as well as a boundary that is not easy to identify; and (3) be distracted by complex scene background. First, Gridmask data augmentation is used to increase the number of labeled images, where potential occlusions are considered by using random information deletion. The augmented images are then fed to the improved encoder, where the traditional $3 \times 3$ convolution is replaced with dynamic convolution to boost the model ability in extracting the fire feature with varying size and shape. The feature refinement module refines the encoder feature by integrating the channel and spatial attention and increase the robustness to complex scene background in remote sensing images. Extensive experiments conducted on two publicly available datasets show that the proposed method achieves superior performance for forest fire segmentation compared with state-of-the-art semi-supervised segmentation networks. Our method also yields a

competitive result in comparison with fully supervised semantic segmentation, showing that the use of unlabeled data allows the model to automatically extract the meaningful information of the fire, effectively mitigating the influence of artificial annotation noise.

In this paper, we explore the application of semi-supervised learning to remote-sensing-based forest monitoring, which provides an efficient solution for designing a model with limited labeled data. In the future, we will continue to explore the semi-supervised learning method and increase the accuracy of the model.

**Author Contributions:** J.W. contributed to writing the draft, conducting experiments and visualizing the results; X.F. contributed to revising the draft and methodology; X.Y. and T.T. contributed to reviewing and editing the draft; Y.W. contributed to processing the data. All authors have read and agreed to the published version of the manuscript.

**Funding:** This work was supported by the joint fund of Science & Technology Department of Liaoning Province and State Key Laboratory of Robotics (No. 2020-KF-22-04), the National Natural Science Foundation of China (No. 61902187), and the Postgraduate Research & Practice Innovation Program of Jiangsu Province (KYCX22_1105).

**Data Availability Statement:** The Flame dataset is available at https://ieee-dataport.org/open-access/flame-dataset-aerial-imagery-pile-burn-detection-using-drones-uavs. The Corsican dataset is available at http://cfdb.univ-corse.fr/index.php?menu=1 (Accessed on 19 January 2022).

**Conflicts of Interest:** The authors declare no conflict of interest. The founding sponsors had no role in the design of the study; in the collection, analyses, or interpretation of data; in the writing of the manuscript, or in the decision to publish the results.

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
