# Peer review of "Semi-Supervised Learning for Forest Fire Segmentation Using UAV Imagery"

_forests, doi:10.3390/f13101573_

Round 1
Reviewer 1 Report
Please find the comments as follows for the minor revision.
1) Line 63-79: Please refine the motivation to make it concise and clear.
2) Figure 4: It will be better to mark detailed information for this figure, e.g., the scale values.
3) Figure 12, 13, 14: Please make them clear enough to see the test within the figures.
4) Please make sure there is no indent: Line 325, 369, and 383.
Reviewer 2 Report
1. The compared networks are too few to reach the advanced level.
2. Equations (4), (5), (6), (7) are not fully explained, each variable or function should have a clear explanation.
3. Too few evaluation metrics are used, IOU is not fully used to evaluate the results, such as the common Precision, Recall, F1-Score are not used.
4. The authors are the segmentation of complex scenes, but the listed segmentation images can not bring out the complex scenes. The authors should put challenging images in Figure 9 to reflect the complexity of the scene.
6. The authors should put challenging images in the segmentation result (Figure 11), such as flame features with different sizes and shapes.
7. In Figure 4 the authors show the masking effect of the values of (r,d) in different cases. The authors should explain Figure 4 so that the reader can understand the impact of these three scales.
Round 2
Reviewer 2 Report
The authors have addressed all the queries.